# Spatiotemporal Variations of Radon Concentration in the Atmosphere of Zhijindong Cave (China)

**Xu Weng** [1,2], **Weijun Luo** [1,3,4,*] , **Yanwei Wang** [1,4,*], **Guangneng Zeng** [4,5] **and Shijie Wang** [1,4]

1   State Key Laboratory of Environmental Geochemistry, Institute of Geochemistry, Chinese Academy of Sciences, Guiyang 550081, China; wengxu@mail.gyig.ac.cn (X.W.); wangshijie@mail.gyig.ac.cn (S.W.)
2   University of Chinese Academy of Sciences, Beijing 100049, China
3   College Rural Revitalization Research Center of Guizhou, Anshun 561000, China
4   Puding Karst Ecosystem Research Station, Chinese Academy of Sciences, Anshun 562100, China; zgl880713@126.com
5   School of Eco-Environmental Engineering, Guizhou Minzu University, Guiyang 550025, China
*   Correspondence: luoweijun@vip.gyig.ac.cn (W.L.); wangyanwei@mail.gyig.ac.cn (Y.W.)

**Abstract:** Ensuring high air quality in the atmosphere of Zhijindong Cave is essential, for it is one of the most scenic in Asia and has received millions of tourists each year. Radon, as the most important radioactive carcinogen, is a priority and has been measured since just after its opening. However, an artificial exit was opened in 2002, and it is still unclear what the influence of that has been on the radon concentration in the cave atmosphere. In this study, we use RAD7 to monitor the spatiotemporal variations of radon concentration in the atmosphere of Zhijindong Cave for a whole year. The results show that radon concentration is generally higher in the hot season and lower in the cold season, and both with a distinct spatial differences. The highest measured radon concentration is 1691 Bq/m$^3$, which is lower compared with the previous study. The reduced radon concentration could be caused by the strengthened cave ventilation due to the artificial exit. The temporal variation of radon concentration is related to the outside temperature change, while the spatial variation is mostly related to the different cave layers. The effective dose is negligible for tourists, but can be as high as 9.7 mSv for tour guides and 22.6 mSv for photographers.

**Keywords:** Zhijindong cave; radon; cave ventilation; radiation dose; health risk

## 1. Introduction

Radon ($^{222}$Rn) is the decay product of radium ($^{226}$Ra) in the uranium ($^{238}$U) decay chain. It is a colorless and tasteless radioactive gas with a half-life of 3.8 days. After radon and its daughters are inhaled by human beings, $\alpha$ particles will be released in the decay process, causing damage to human respiratory tract and lungs. Studies have shown that ionizing radiation with high radon exposure can cause lung cancer. Radon is the biggest carcinogenic factor besides smoking for regular smokers, and for non-smokers, radon is the biggest carcinogenic factor for lung cancer [1].

Radon in the natural environment mainly comes from the release of rock and soil, and the concentration difference is obvious in different regions (different background values of uranium and radium) and different environments (outdoor, basement, indoor, cave, and mine). Especially, local radiation which is much higher than natural background radon radiation (1.2 mSv/a) has been found in Yangjiang (China), Guarapari (Brazil), Kerala (India), and Lamsal (Iran) [2–4]. Radon concentration in the open atmosphere generally ranges from a few Bq/m$^3$ to more than a dozen Bq/m$^3$ [5,6], while in indoors and basements can reach thousands of Bq/m$^3$ [7]. Radon concentrations can reach hundreds of thousands of Bq/m$^3$ in some cave environments [8,9]. Although the content of uranium in limestone is low, carbonates are prone to uranium enrichment during weathering [10], which can resulting in higher radon concentrations in caves in many karst regions of

the world [11–14].The concentration of radon in karst cave air depends on the complex interaction of several different factors, including the internal and external temperature difference, the mixing degree of external air and cave air, the internal humidity of the cave, precipitation and its infiltration into the cave environment, the content of $^{226}$Ra in rock strata and cave sediments, the porosity of rock and cave sediments, and the volume and shape of the cave [15,16]. Among them, the cave ventilation caused by meteorological conditions is considered to be the biggest influencing factor [9].

Cave radon research began in the 1970s and 1980s, and radon monitoring has been carried out in a large number of tourist caves [17–23]. Zhijindong Cave is one of the most scenic in Asia and has received millions of tourists each year. There are many observational studies on Zhijindong Cave, including $CO_2$ [24], microorganisms [25], and drip water [26]. Radon as the most important radioactive carcinogen has been measured since just after its opening [27]. However, an artificial exit was opened in 2002 due to the increasing number of tourists, and the influence of this action on the radon concentration in the cave atmosphere it is still unclear. The ventilation of the cave may have changed greatly and the associated radon concentration could be changed too, so it is necessary to update the radon measurement in Zhijindong Cave. In this study, we use RAD7 to monitor the spatiotemporal variation of radon concentration in the atmosphere of Zhijindong Cave for a whole year. Our goal is to comprehensively monitor the radon concentration of various tourist attractions in Zhijindong Cave and evaluate the potential health risk based on the calculated effective dose, so as to provide guidance for the follow-up tourism development of Zhijindong Cave.

## 2. Materials and Methods

### 2.1. Site Description

Zhijindong Cave (26°38′31″ N~26°52′35″ N, 105°44′42″ E~106°11′38″ E) is located in Zhijin County in the karst area of southwest China. This area belongs to the subtropical region of China, and the climate type is the subtropical monsoon climate, the annual average temperature is 14.7 °C, the annual average precipitation is about 1400–1500 mm, the local vegetation type is mainly subtropical evergreen broad-leaved forest [28]. The cave is developed in the limestone strata of Huangchunba member (T1y2) and Yongningzhen formation (T1yn) in Yelang formation (T1y) of Lower Triassic [29]. It is composed of two main caves and four branch caves with a total length of 12.1 km and a total of 47 halls, which can be divided into four layers from bottom to top (Figures 1 and 2). The first layer is in the bottom of the cave, connecting to the underground river, and the second layer is the unopened section. The open section is 3.4 km long, which is mainly located on the third and fourth floors, with a total area about 780,000 m$^2$ and a total volume about 6 million m$^3$. The highest part of the tunnel is 78 m, the widest part is 175 m, and the relative height difference is 150 m.

Zhijindong Cave was discovered in 1980 and opened to tourists in 1985. The secondary chemical deposition landscape in the cave was rich, and the typical stalactite deposits were well developed. Zhijindong Cave has the reputation of "King of Karst Caves in China" and has attracting millions of tourists to visit each year. In Zhijindong Cave, cement artificial trails are laid and a certain amount of illumination lights and landscape lights are installed. The artificial tunnel dug at the end of the cave in 2002 may increase the natural ventilation of the cave. No mechanical ventilation was used in Zhijindong Cave.

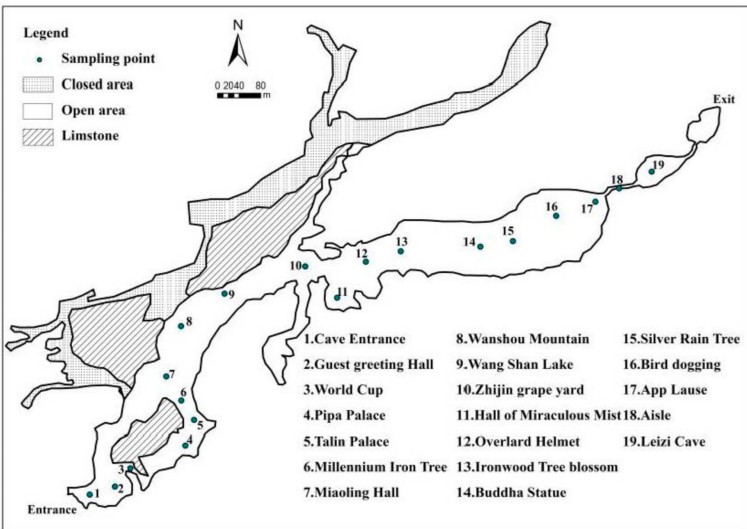

**Figure 1.** Plane map of Zhijindong Cave and the distribution of sampling point. (Revised from references [30]).

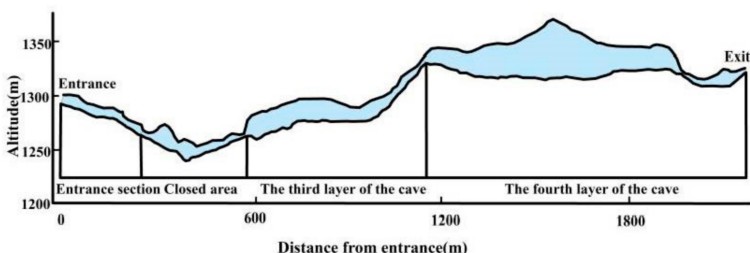

**Figure 2.** Profile of Zhijindong cave (Revised from references [30]).

*2.2. Radon Monitor Method*

Radon measurements in Zhijindong cave are mainly done using a RAD7 electronic radon detector produced by the Durridge Company (Billerica, MA, USA). The instrument provides three measurement modes: sniff mode, normal mode, and auto mode, which can measure the radon concentration in the air continuously and directly. The α particles produced during the decay of radon and its daughters are received by the detector under high pressure. The instrument detects the radon concentration by detecting α particles, ranging from 3.7 Bq/m$^3$ to millions of Bq/m$^3$. When RAD7 is used for radon survey, the counting fluctuation and instrument uncertainty caused by radioactive decay are the main sources of measurement uncertainty. The research on RAD7 measurement accuracy shows that when sniff mode is selected for radon measurement, it can give more reliable measurement values after 12 min, and when other modes are selected for continuous monitoring, it can give more reliable measurement values after 120 min [31].

The monitoring of radon in Zhijindong Cave is mainly carried out by instantaneous monitoring and continuous monitoring. During the period from July 2020 to June 2021, the sniff model (a measurement once every five minutes) was used to measure radon concentration three times at each representative scenic site to explore the spatial and monthly changes of radon concentration in Zhijindong Cave. Generally, we choose a sunny day each month to conduct the monitor work which should be representativeness of the seasonal temperature variation. Another instrument was installed at monitoring site # 12 to select the normal mode (measured once an hour) for continuous monitoring for two months to explore the diurnal changes of radon concentration at a single site in Zhijindong cave. The meteorological data was derived from Zhijin County Meteorological Monitoring Station.

Radon concentration threshold is used directly to control radon levels in homes and most workplaces. However, it is necessary to calculate the effective dose to quantify the potential health risk. Currently, the International Commission on Radiological Protection (ICRP) uses the dose conversion convention to calculate effective dose per unit exposure to radon and its progeny. Based on the recommendations of the latest publication of ICRP, we use the dose conversion factor of approximately 20 mSvWLM$^{-1}$ which is recommended for workers in tourist caves [32]. And to calculate the WLM using the formula as:

$$\text{WLM} = \frac{\sum(C_{Rn}Ft)}{3700\,\text{Bq}/\text{m}^3 \times 170\,\text{h}} \tag{1}$$

where $C_{Rn}$ is the measured radon concentration, in Bq/m$^3$. $F$ is the equilibrium factor, which for a tourism cave is 0.4. $t$ is residence time in the cave, in h.

## 3. Results

### 3.1. Temporal Variation of Radon Concentration

The radon concentration shows obvious seasonal variation with higher values in hot seasons and lower values in cold seasons in the atmosphere of Zhijindong Cave (Figure 3). The overall annual average radon concentration is 509 Bq/m$^3$, and the highest value is 1692 Bq/m$^3$, which is lower than the highest radon concentration measured in Zhijindong Cave (2191 Bq/m$^3$) by an earlier study [27]. Compared with some karst tourist caves in China, the radon concentration of Zhijindong Cave is lower than that of Xueyu cave (4271.4 Bq/m$^3$) [22] and is close to that of caves in Lantian [15]. Compared with some other karst tourist caves in the world, the radon concentration in Zhijindong Cave is higher than caves in Romania [33], lower than the Važecká cave (1300–42,200 Bq/m$^3$) and the Altamira cave (3562 Bq/m$^3$) [11,16].

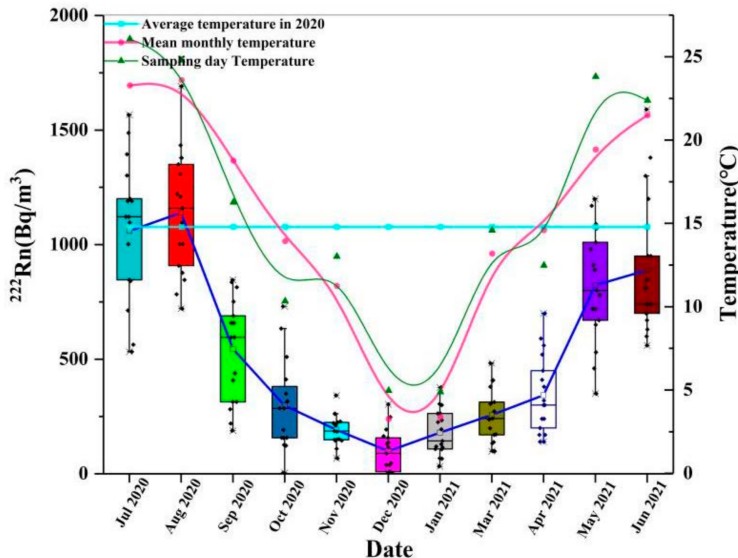

**Figure 3.** Seasonal variation of radon concentration and temperature.

The temporal variation of radon concentration is closely related to the temperature difference between the outside atmosphere and the cave atmosphere (Figure 4). The radon concentration will increase when the temperature of the outside atmosphere is higher than the cave air temperature, and will decline when the temperature of the outside atmosphere is lower than the cave air temperature. And this phenomenon could be caused by cave ventilation process through chimney effect [9]. The radon concentration in the atmosphere at monitor site #12 shows little variation on the hourly scale (Figure 5). This proves that our monitor schedule to get spatial variation of radon concentration with one mobile RAD7

is representative, because the time variation of radon concentration is little with limited time duration.

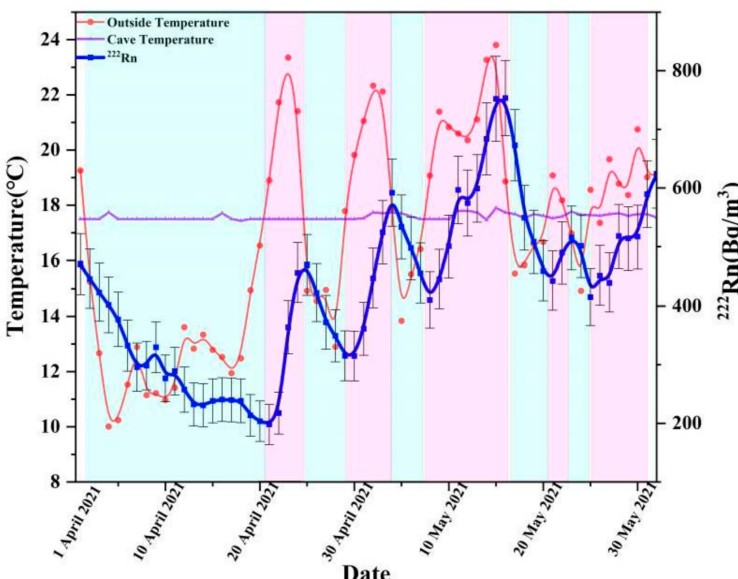

**Figure 4.** Relationship between radon concentration variation and temperature difference at the monitor site #12.

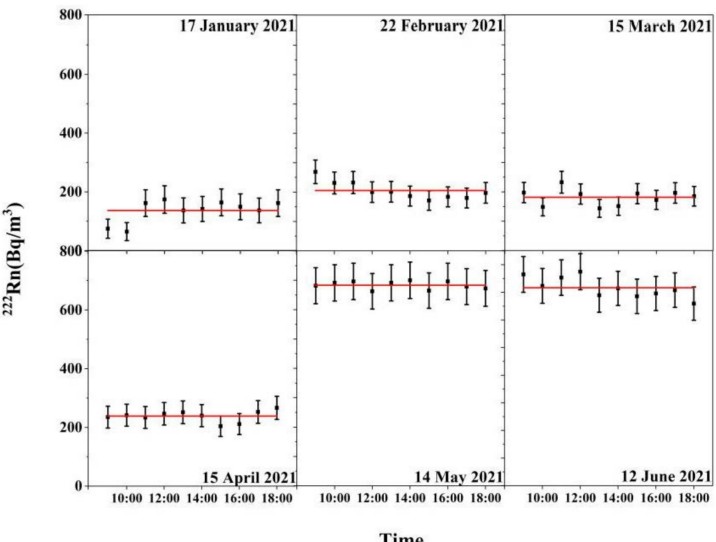

**Figure 5.** Diurnal variation of radon concentration at the monitor site #12.

### 3.2. Spatial Variation of Radon Concentration

There is a distinctly different spatial variation of radon concentration between hot seasons and cold seasons in the atmosphere of Zhijindong Cave (Figure 6). Generally, the radon concentration is higher in the deep layers of the cave during the hot seasons, while in the cold seasons the higher radon concentration is in the middle part. Because the tourist route is long and complex in Zhijindong Cave. The correlation between the spatial parameters (length, area, volume, depth) and the variation of radon concentration are complex in Zhijindong Cave. We find that there is no obvious correlation between the size of the cavern space and the variation of radon concentration, while the depth of the cavity has great influence on the spatial variation of radon concentration.

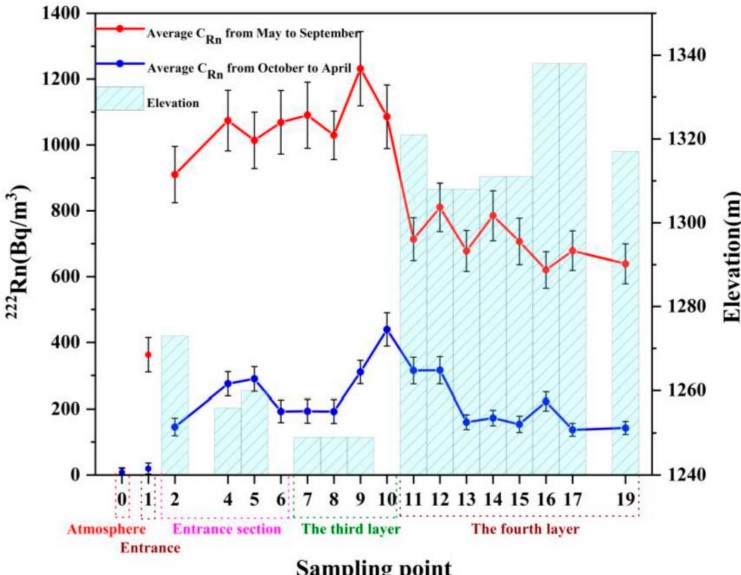

**Figure 6.** Spatial variation of radon concentration in Zhijindong Cave.

The complex spatial variation could be related to the cave ventilation process. From May to September, the air in the vadose zone including cave atmosphere is colder than external air, so the overall ventilation of is downward which brings the high background air through the cave entrance and exit. Therefore, radon concentration in the entrance and exit are much higher than atmosphere background. And the radon concentration in entrance section and the third layer is higher than the fourth layer, which could be caused by thicker vadose zone above the cave roof. From October to April, the temperature of inside air is higher than the outside air, so the overall ventilation is upward which brings outside air into cave space, so the radon concentration at the entrance is close to atmosphere background. Because the elevation of the exit is relatively high, so the cave air flow out the exit and the radon concentration is close to the cave signal.

## 4. Discussion

The overall annual average radon concentration in Zhijindong Cave is higher than 300 Bq/m$^3$ recommended by ICRP for indoor environment and lower than 1000 Bq/m$^3$ for working environment [34], but the radon concentration during hot seasons in many scenic spots is higher than 1000 Bq/m$^3$, so there will be potential health risk if one stays for a long time in Zhijindong Cave. Different types of staff (tour guides, photographers, environmentalists, etc.) have different working locations and time duration, so the calculation of effective dose should be divided according to regions, time and population. The people are mainly divided into four categories in Zhijindong Cave: tourists, cleaning staff, tour guides and scenic spot photographers (uninterrupted throughout the year, working for 2–3 years). It takes about 1.5 h for most tourists to finish their visit to the cave. The tour guide works in the cave 3 to 4 times a day, and the exposure time is 6 h at most. Cleaning staff and photographers stay in the cave for a long time, and their exposure time is 9 h at most for a whole day. Tourists, tour guides and cleaning staff stay irregularly at all points in the cave. Therefore, we use the highest average radon concentration of the whole cave in August and the lowest average radon concentration of the whole cave in December to calculate the possible range of effective dose. The exposure dose of tour guides and cleaners is calculated by the annual average radon concentration. The photographer mainly stayed at monitoring points #12 and #15, and the annual average radon concentration at these two sites were used to calculate the effective dose. The maximum personal effective dose of radon in Zhijindong Cave is 22.6 mSv for photographers, but the effective dose to tourists are much lower than the background radiation dose of 2.4 mSv/a (Table 1) [2].

The effective dose of workers is close to the dose limit recommended by ICRP, which is an average of 20 mSv per year for 5 years, and the dose in any year does not exceed 50 mSv [35]. Therefore, under this standard, the working time of staff in Zhijindong Cave should be managed to reduce the potential health risk.

**Table 1.** Summary of average radon concentration and typical effective dose.

| Classify | Exposure Time (h) | Average Radon Concentration (Bq/m³) | Dose (mSv) |
|---|---|---|---|
| Tourists | 1.5 | August: 1126<br>December: 99 | 0.02<br>0.002 |
| Tour guide | 1500 | 509 | 9.7 |
| Cleaner | 2250 | 509 | 14.6 |
| Photographer | 3285 | Site #15: 405<br>Site #12: 541 | 16.9<br>22.6 |

Ventilation is the key process to influence gas exchanges in the cave environment. $CO_2$ is often used as a tracer gas for the study of cave ventilation processes [36], but in tourist caves, it is often disturbed by tourists' breathing and cannot accurately indicate the cave ventilation. The study on the change of $CO_2$ in Zhijindong Cave in the past 20 years shows that since 1989, the $CO_2$ concentration in Zhijindong Cave has generally increased [37]. Although a second artificial exit was opened in 2002 and the ventilation of the cave was improved, the $CO_2$ concentration continues to increase due to the increase of tourists. However, the highest radon concentration measured in our study is lower than that measured by an earlier study [27], and this information could be used to distinguish the impact of human activity and natural ventilation in further studies.

## 5. Conclusions

This study updates the spatiotemporal variations of radon concentration in the atmosphere of Zhijindong Cave. The monthly monitoring results show that the radon concentration is generally higher in summer and lower in winter, and the radon concentration in the deep layer of the cave is higher than in the upper layer. The temporal variation is mainly affected by the cave ventilation which is dominated by the external temperature difference. The overall radon concentration has been reduced due to the opening of an artificial exit, but the effective dose for long-term workers is still close to the recommended threshold by ICRP and their work time should be reduced appropriately. In future studies, long-term continuous radon monitoring should be conducted to calculate the effective dose more accurately.

**Author Contributions:** Conceptualization, W.L., G.Z. and Y.W.; methodology, Y.W. and X.W.; software, X.W.; validation, W.L.; formal analysis, X.W. and Y.W.; investigation, X.W. and Y.W.; resources, W.L.; data curation, X.W.; writing—original draft preparation, X.W.; writing—review and editing, X.W., Y.W., G.Z. and W.L.; visualization, X.W.; supervision, W.L. and Y.W.; project administration, W.L.; funding acquisition, W.L. and S.W. All authors have read and agreed to the published version of the manuscript.

**Funding:** This research was funded by the Strategic Priority Research Program of the Chinese Academy of Sciences, grant number XDB40020200; the National Natural Science Foundation of China, grant number 41673121 and 41663015; Guizhou Science and Technology Cooperation Basic Project (2020)1Y188.

**Institutional Review Board Statement:** Not applicable.

**Informed Consent Statement:** Not applicable.

**Data Availability Statement:** Not applicable.

**Acknowledgments:** We want to thank the editor and anonymous reviewers for their valuable comments and suggestions. We also want to thank the Zhijindong Cave National Geopark Administration for their permission and support, especially Zhiyong Liu and Yongguo Dai for their help.

**Conflicts of Interest:** The authors declare no conflict of interest.

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
