# Peer review of "Spatiotemporal Variations of Radon Concentration in the Atmosphere of Zhijindong Cave (China)"

_atmosphere, doi:10.3390/atmos12080967_

Round 1

Reviewer 1 Report

[General comments]

This paper discussed the spatiotemporal variation in atmospheric radon concentrations in the Zhijindong cave (China), based on short-term monitoring. The reviewer is concerned about the representativeness of radon concentrations obtained in this study; without the representativeness, effective doses calculated would sometimes be quite misleading. This point should be reinforced by additional explanation or data before publication in the journal.

[Specific comments]

Line 36: Bq/m --> Bq/m3

Line 45: International radiation Commission --> International Commission on Radiological Protection

Line 50-52 (Generally, the harm of low dose radiation … linear thresholdless model): This expression is not correct; the deterministic effect is not related to the LNT model. Please revise it carefully.

Line 41-53: This paragraph includes a detailed explanation of radiation risk, which is not necessarily working as an introduction to the present paper. Rather, information on earlier cave studies might be more useful.

Line 69 (2. Materials and Methods): Please add some information on natural and/or mechanical ventilation in the cave.

Line 90 (Figure 1): The image is too small, and its resolution is too low. Thus, the explanation on the layers, given at Line 79-81, is difficult for the reviewer to understand.

Line 110: balance factor --> equilibrium factor.

Line 110: The reviewer could not understand what the sentence “not considering unattached particles” means. In general, factor F includes both attached and unattached fractions. Also, is the value of 0.4 a measured or assumed/cited value?

Line 111-112: Please explain how to assume 13.4E-6 mSv/Bq/h/m3 for tour guides and staff and 6.7E-6 mSv/Bq/h/m3 for the public. As far as the reviewer saw the ICRP Publ. 137 (Ref [35] of the present paper), a dose coefficient of 1.5E-5 mSv/Bq/h/m3 was recommended for tour cave staff.

Line 130-131: Please explain what the sentence “good relation” means here; in addition, what kind of scientific meaning the “good relation” represent? How was the correlation coefficient between the two parameters? If the authors insist on the relationship between radon concentration and outside-inside temperature difference, then the differential temperature should be exhibited in Figure 3.

Line 130-131: There may be a time lag between radon concentration and outside-inside temperature difference. Is it not necessary to refer to and discuss this time lag?

Line 134-135: The reviewer could not which part of Figure 3 shows the stable radon concentration and stable temperature difference. Please state it explicitly.

Line 137: Please mention, in an appropriate place, how to decide the monitoring schedule in this study. Were there some criteria (e.g., a sunny day not after rainy days)?

Line 140 (Figure 4): Why did not the authors show the data for July 2020 to December 2020? Even if they are similar to Figure 4, then the authors should mention so in the text. Also, please correct “1/17/2021” to 17/1/2021 at the upper left panel.

Line 161-162 (the overall ventilation is upward which brings outside air into cave space): Is this correct? Just a simple concern.

Line 188-189 (Therefore, there is no health risk…): This conclusion (i.e., no health risk) is quite misleading and must be revised carefully. Please confirm that if a dose is not zero, then the risk is also not zero from the viewpoint of radiation protection.

Line 191-215: There seems to be a sentenced explaining that this cave is safe for tourists and staff. However, this paragraph includes some controversial issues and also is not reasonably used for the deduction of the conclusion. Maybe, discussion from other aspects (e.g., time management) seems more reasonable and preferable.

Line 216: The representativeness of this monitoring and calculated doses must be stated. Namely, how was the variation in radon concentrations for other days, while only a single day was used every month for the monitoring? How is the dependence of radon concentrations on meteorology or climate? How appropriate was the factor F of 0.4 for this cave?

Author Response

Response to Reviewer 1 Comments

Point 1: This paper discussed the spatiotemporal variation in atmospheric radon concentrations in the Zhijindong cave (China), based on short-term monitoring. The reviewer is concerned about the representativeness of radon concentrations obtained in this study; without the representativeness, effective doses calculated would sometimes be quite misleading. This point should be reinforced by additional explanation or data before publication in the journal. 

Response 1: Thanks a lot for reviewing our manuscript. Indeed, long-term continuous monitoring of radon concentration is warranted to the representativeness of radon concentrations as well as to the calculated effective doses. In this study, we measured radon concentration along the travel route once each month for a whole year in Zhijindong Cave to ensure spatial and seasonal representativeness. Additionally, the two months continuous radon concentration data at monitor site #12 could be used to verify its dynamic change with temperature variation. Also, the diurnal variation characteristics of radon concentration can tell us daily representativeness. So the results obtained in this study should be close to the real situation. Your comments and suggestions are valuable and the manuscript has been improved accordingly.

Point 2: Line 36: Bq/m → Bq/m3

Response 2: Thanks, the mistake will be corrected in the revised manuscript.

Point 3: Line 45: International radiation Commission → International Commission on Radiological Protection

Response 3: Thanks, the mistake will be corrected in the revised manuscript.

Point 4: Line 50-52 (Generally, the harm of low dose radiation … linear thresholdless model): This expression is not correct; the deterministic effect is not related to the LNT model. Please revise it carefully.

Response 4: Thanks, the expression will be corrected in the revised manuscript.

Point 5: Line 41-53: This paragraph includes a detailed explanation of radiation risk, which is not necessarily working as an introduction to the present paper. Rather, information on earlier cave studies might be more useful.

Response 5: Thanks, the related introduction information will be improved in the revised manuscript.

Point 6: Line 69 (2. Materials and Methods): Please add some information on natural and/or mechanical ventilation in the cave.

Response 6: Thanks, the related information will be added in the revised manuscript.

Point 7: Line 90 (Figure 1): The image is too small, and its resolution is too low. Thus, the explanation on the layers, given at Line 79-81, is difficult for the reviewer to understand.

Response 7: Thanks, the detail of the image will be improved in the revised manuscript.

Point 8: Line 110: balance factor --> equilibrium factor.

Response 8: Thanks, the expression will be improved in the revised manuscript.

Point 9: Line 110: The reviewer could not understand what the sentence “not considering unattached particles” means. In general, factor F includes both attached and unattached fractions. Also, is the value of 0.4 a measured or assumed/cited value?

Response 9: Thanks, the sentence is misleading and will be removed in the revised manuscript, and the F value of 0.4 is a cited value in this study.

Point 10: Line 111-112: Please explain how to assume 13.4E-6 mSv/Bq/h/m3 for tour guides and staff and 6.7E-6 mSv/Bq/h/m3 for the public. As far as the reviewer saw the ICRP Publ. 137 (Ref [35] of the present paper), a dose coefficient of 1.5E-5 mSv/Bq/h/m3 was recommended for tour cave staff.

Response 10: Indeed, a dose coefficient of 1.5E-5 mSv/Bq/h/m3 (24 mSv WLM-1) was recommended for tourist cave in Table 12.7 of Ref [35]. However, in considering the consistency between dose coefficients obtained by dosimetric calculations and by epidemiological comparisons,  the Commission recommends a dose coefficient of 20 mSv WLM-1 (1.3E-5 mSv/Bq/h/m3) to be more appropriate for workers in tourist caves on page 448 in (A8) of Ref [35].

Point 11: Line 130-131: Please explain what the sentence “good relation” means here; in addition, what kind of scientific meaning the “good relation” represent? How was the correlation coefficient between the two parameters? If the authors insist on the relationship between radon concentration and outside-inside temperature difference, then the differential temperature should be exhibited in Figure 3.

Response 11: Thanks, the sentence is misleading and will be improved in the revised manuscript. What we want to express is that there is a close connection between the outside-inside temperature difference and the variation of radon concentration.

Point 12: Line 130-131: There may be a time lag between radon concentration and outside-inside temperature difference. Is it not necessary to refer to and discuss this time lag?

Response 12: Indeed, there may be a time lag between radon concentration and outside-inside temperature difference. But the continuous monitoring data show this time lag is negligible in our study.

Point 13: Line 134-135: The reviewer could not which part of Figure 3 shows the stable radon concentration and stable temperature difference. Please state it explicitly.

Response 13: Thanks, the sentence is misleading and will be removed in the revised manuscript.

Point 14: Line 137: Please mention, in an appropriate place, how to decide the monitoring schedule in this study. Were there some criteria (e.g., a sunny day not after rainy days)?

Response 14: Thanks, the criteria of the monitoring schedule will be added in the revised manuscript. Generally, we choose a sunny day each month to better represent the monthly average temperature condition.

Point 15: Line 140 (Figure 4): Why did not the authors show the data for July 2020 to December 2020? Even if they are similar to Figure 4, then the authors should mention so in the text. Also, please correct “1/17/2021” to 17/1/2021 at the upper left panel.

Response 15: Thanks, the mistake will be corrected in the revised manuscript. We show the diurnal radon variation to verify the representativeness of our monitor schedule. The diurnal variation pattern should be similar for July 2020 to December 2020, although no data was available in this study.

Point 16: Line 161-162 (the overall ventilation is upward which brings outside air into cave space): Is this correct? Just a simple concern.

Response 16: The low radon concentration should be caused by the mix of atmosphere air. The ventilation process was controlled by the chimney effect in Zhijindong cave, so the overall ventilation should be upward during wintertime. More information can be found in Ref [9].

Point 17: Line 188-189 (Therefore, there is no health risk…): This conclusion (i.e., no health risk) is quite misleading and must be revised carefully. Please confirm that if a dose is not zero, then the risk is also not zero from the viewpoint of radiation protection.

Response 17: Thanks, the expression will be improved in the revised manuscript. What we want to express is that the calculated effective dose is within the limits of ICRP recommendations.

Point 18: Line 191-215: There seems to be a sentenced explaining that this cave is safe for tourists and staff. However, this paragraph includes some controversial issues and also is not reasonably used for the deduction of the conclusion. Maybe, discussion from other aspects (e.g., time management) seems more reasonable and preferable.

Response 18: Thanks, this part of discussion will be substituted in the revised manuscript.

Point 19: Line 216: The representativeness of this monitoring and calculated doses must be stated. Namely, how was the variation in radon concentrations for other days, while only a single day was used every month for the monitoring? How is the dependence of radon concentrations on meteorology or climate? How appropriate was the factor F of 0.4 for this cave?

Response 19: Thanks, this part of conclusion will be improved in the revised manuscript. Indeed, the representativeness of this monitoring is certainly limited. But the general seasonal pattern of radon concentration in Zhijindong cave as well as its relationship with outside-inside temperature difference is obvious in our study. In future studies, long-term continuous radon monitor, as well as site-specific F value, should be conducted to calculate effective dose more accurately.

Response to Reviewer 2 Comments

Point 1: The topic is of interest, however the paper must be significantly improved before it can be considered acceptable for publication. Some specific points: 

Response 1: Thanks a lot for reviewing our manuscript. Your comments and suggestions are valuable and the manuscript has been improved accordingly.

Point 2: The authors too much speculate about low doses risk without standing on their own evidence. However, for chronic radon exposure there is a direct epidemiological evidence for the risk - suggest to refer to WHO Handbook on Indoor Radon (2009), instead of speculations at lines 48-53.

Response 2: Thanks, the expression related to the controversial low dose risk will be removed in the revised manuscript. We will focus on our own data and quantify the effective dose for workers and tourists in Zhijindong cave.

Point 3: l. 45-46: Please, spell out ICRP correctly: International Commision on Radiological Protection.

Response 3: Thanks, the mistake will be corrected in the revised manuscript.

Point 4: lines 184, 186: please use "effective dose", not the wrong term "exposure dose"

Response 4: Thanks, the mistake will be corrected in the revised manuscript.

Point 5: Suggest to cancel-out the sentence that starts on l. 188.

Response 5: Thanks, the sentence is misleading and will be improved in the revised manuscript.

Point 6: Suggest to cancel-out the text between 197-215. This is rather speculative consideration, not supported at all by the findings of the manuscript itself.

Response 6: Thanks, in the original manuscript, we want to highlight the possible beneficial health effect related to low dose radiation based on the hormetic effect, but this is surely beyond the scope of this article. And this part of discussion will be removed in the revised manuscript. We will focus on our own data and give advice to limit the effective dose within the ICRP recommendation.

Reviewer 2 Report

The topic is of interest, however the paper must be significantly improved before it can be considered acceptable for publication. Some specific points:

  • The authors too much speculate about low doses risk without standing on their own evidence. However, for chronic radon exposure there is a direct epidemiological evidence for the risk - suggest to refer to WHO Handbook on Indoor Radon (2009), instead of speculations at lines 48-53.
  • l. 45-46: Please, spell out ICRP correctly: International Commision on Radiological Protection.
  • The authors should address the issue of the uncertainty in measurements by RAD 7 in Sniff mode - refer e.g. to Pressyanov et al. AIP Conf. Proc. 1607, 24 (2014).
  • lines 184, 186: please use "effective dose", not the wrong term "exposure dose"
  • Suggest to cancel-out the sentence that starts on l. 188.
  • Suggest to cancel-out the text between 197-215. This is rather speculative consideration, not supported at all by the findings of the manuscript itself.

Author Response

Thanks a lot for reviewing our manuscript. Your comments and suggestions are valuable and the manuscript has been improved accordingly.

Round 2

Reviewer 2 Report

The paper is improved and may be published.